# Process-based Modeling of Solar-induced Chlorophyll Fluorescence with VISIT-SIF version 1.0

Tatsuya Miyauchi[1,2]*, Makoto Saito[2]*, Hibiki M. Noda[2], Akihiko Ito[3], Tomomichi Kato[1], and Tsuneo Matsunaga[2]

[1]Research Faculty of Agriculture, Hokkaido University, N9W9 Sapporo, Hokkaido 060-8589, Japan
[2]Earth System Division, National Institute for Environmental Studies, 16-2 Onogawa, Tsukuba, Ibaraki 305-8506, Japan
[3]Graduate School of Agricultural and Life Sciences, University of Tokyo, 1-1-1 Yayoi, Bunkyo, Tokyo 113-8657, Japan

*Correspondence to*: Tatsuya Miyauchi, Makoto Saito (tmiyauchi@agr.hokudai.ac.jp; saito.makoto@nies.go.jp)

**Abstract.** Satellite retrievals of solar-induced chlorophyll fluorescence (SIF) can provide opportunities to improve our understanding of terrestrial ecosystem dynamics and the carbon cycle at the global scale. Here, we present a new biogeochemical process-based carbon and nitrogen cycle model for representing SIF retrievals (VISIT-SIF version 1.0) acquired by the Greenhouse gases Observing SATellite (GOSAT) with an hourly time step and a spatial resolution of approximately $0.31 \times 0.31$ degrees. Implementation of radiation transfer models (RTMs) helps to address the interaction of chlorophyll fluorescence with vegetation and atmosphere. However, the computation of RTMs becomes more time-consuming, which can make it impractical in application to satellite observations with larger data volumes. This study resolves this issue by parameterizing the radiative transfer processes and its geometric relationships. This approach enables ease of implementation of VISIT-SIF for simulating satellite SIF retrievals even for the satellites having off-nadir observation angles. With an initial seven years of data (2009-2015), our model simulations showed a consistent global mean value of $0.51 \pm 0.39$, with GOSAT SIF retrievals of $0.46 \pm 0.42$ mW m$^{-2}$ sr$^{-1}$ nm$^{-1}$; the root-mean-square error was 0.29 mW m$^{-2}$ sr$^{-1}$ nm$^{-1}$. We also found that the mean seasonal variability in the simulated SIFs was mostly consistent with the GOSAT SIF retrievals at the subcontinental scale. However, the simulated results indicated less sensitivity to water stress in the late dry season in arid and semiarid regions relative to that of the GOSAT SIF retrievals, which is consistent with the findings of previous studies using multiple biogeochemical process-based models. This comparison suggested that there is a critical need to improve our knowledge of SIF variability and biophysical processes in such regions.

## 1 Introduction

Carbon fixation by photosynthesis is a fundamental process for carbon cycling in terrestrial ecosystems (Beer et al., 2010). In the photosynthetic process, solar energy absorbed by chlorophylls is mainly dissipated through three pathways: photochemistry at the photosynthetic reaction center, nonradiative energy dissipation into heat, and reemission as a photon of fluorescence (Porcar-Castell et al., 2014). While most of the absorbed solar energy is utilized for photochemistry under light-limited conditions, a small amount of energy (approximately 1-2% of the total absorbed radiation energy) is reemitted as chlorophyll fluorescence in the visible (VIS) to near-infrared spectrum between 640-800 nm (Maxwell and Johnson, 2000). When photosynthesis is light saturated or restricted by environmental stress, the energy flow through heat dissipation increases to prevent damage to the photosynthetic system due to the accumulation of excess energy. The amount of energy consumed by photochemistry and fluorescence decreases with increasing heat dissipation; hence, the quantum yield of photochemistry is correlated with fluorescence and heat dissipation (Flexas et al., 2000). Solar-induced chlorophyll fluorescence (SIF) is the radiation emitted as chlorophyll fluorescence during photosynthesis under natural sunlight conditions. Despite the SIF radiance being weak, recent progress in spectral radiometers with high wavelength resolution has provided capabilities for mapping the global distribution of SIF with satellite observations (Joiner et al., 2011; Frankenberg et al., 2011; Joiner et al. 2013; Sun et al., 2017), as well as those at smaller scales, such as the leaf (Hikosaka and Noda, 2019) and canopy scales (Yang et al., 2015). Current satellite missions commonly quantify SIF emissions from Fraunhofer lines in the oxygen absorption bands between 756 and 759 nm (Oshio et al., 2019) and between 734 and 758 nm (Joiner et al., 2013), with the wavelength corresponding to a spectral peak of SIF at approximately 740 nm that emanates from photosystems I and II (PS I and II). While vegetation indices based on surface reflectance data, such as the normalized difference vegetation index (NDVI; Karlsen et al., 2008, 2014) and enhanced vegetation index (EVI; Wu et al., 2010), have been utilized for describing terrestrial vegetation dynamics, SIF data have attracted attention because of their ability to provide additional information for quantifying the photosynthetic activity of terrestrial vegetation under changing environmental conditions. Indeed, Joiner et al. (2011) and Frankenberg et al. (2011) successfully demonstrated strong correlations between SIF and gross primary production (GPP) for major global land cover types using satellite data acquired by the Greenhouse gases Observing SATellite (GOSAT) (Yokota et al., 2009). In addition, Liu et al. (2018) and Wang et al. (2019) demonstrated the availability of satellite SIF data as a diagnostic indicator for vegetation productivity with a rapid response to underlying environmental stress conditions such as drought. These SIF retrieval characteristics related to the photosynthetic process and dynamic status of vegetation have been implemented in numerous studies for the estimation of GPP, improvement in light use efficiency models (e.g., Guanter et al., 2014; Qiu et al., 2020), identification of environmental stress factors (e.g., Jiao et al., 2019), and adjustment of process-based model parameters (e.g., Norton et al., 2018).

The intensity of SIF is strongly affected by not only physiological processes but also canopy structure, e.g., the leaf area index (LAI) and leaf angle distribution, and the geometric relationships among the incidence angle of the emission to

the sensor, solar azimuth, and orientation of leaves (Porcar-Castell et al., 2014; Zhang et al., 2019). Both the incident solar radiation to the canopy and the SIF emitted from each leaf are reflected, transmitted and absorbed within the canopy, and SIF is emitted upward from the canopy to the sensor. Thus, to incorporate the complex radiative transfer process of SIF, some studies have used radiative transfer models (RTMs) in addition to physiological process models (Koffi et al., 2015; Lee et al., 2015). These studies commonly combined the Soil Canopy Observation of Photosynthesis Energy fluxes (SCOPE; van del Tol et al, 2009, 2014) model for the computation of radiative transfer with respect to the multilayer canopy structure and the geometric relationship. Additionally, in the study by Norton et al. (2019) global SIF retrieved from satellites was used as constraints on biochemical variables in response to photosynthesis based on the data assimilation method, leading to substantial improvements in reducing the uncertainties in global carbon cycle estimates, with a decrease in the uncertainty of global GPP from $\pm 19.0$ PgC $y^{-1}$ to $\pm 5.2$ PgC $y^{-1}$. The exploitation of process-based models, including physiological and radiative transfer processes for satellite-SIF at the global scale, has the potential to promote a better understanding of global carbon cycles, leading to significant advances in the representation of photosynthetic processes.

GOSAT has been operated since the launch in January 2009, and of which SIF retrievals have the longest observation record of any single satellite sensor. Although the primary mission of GOSAT is to measure the column-averaged dry air molar fractions of carbon dioxide ($CO_2$) and methane ($CH_4$) to constrain the global distributions of their sources and sinks and improve understanding of carbon-climate feedbacks, SIF retrievals will provide independent constraints on terrestrial ecosystem carbon dynamics. In general, strict representation of radiative transfer process is required to simulate satellite SIF retrievals, especially for the satellites having off-nadir observation angles due to their complicated geometric relationships among the incidence angle of the emission to the sensor, solar azimuth, and orientation of leaves (Zhang and Zhang, 2023). But at the same time, the resolution of satellite images has kept increasing (e.g., Vicent et al., 2016), and implementation of RTMs for the observations are computationally expensive and prohibitively time-consuming. Hence, a simplified framework of avoiding direct calculation of RTMs can be an alternative and practical approach to simulate efficiently the satellite SIF retrievals. This study aimed to develop a process-based model for representing chlorophyll fluorescence emissions during photosynthesis in major land cover types and a framework for estimating variability in GOSAT SIF, whereby the radiative transfer process from the surface canopy to satellite measurements is adjusted by utilizing a simplified SCOPE model. The implementation of our approach allows for easy extension to other satellite SIF retrievals. Our objective for constructing the model framework is to stimulate the study of terrestrial ecosystem dynamics by improving the formulation of related biophysical processes based on a combination of modeling approaches and GOSAT SIF.

## 2 Methods

 ### 2.1 GOSAT SIF data

We used the SIF data acquired by GOSAT as reference observations for evaluating the model estimates. GOSAT was launched in January 2009 on a sun-synchronous orbit at an altitude of 666 km with a 3-day revisit cycle and a descending node at approximately 13:00 local time. GOSAT was dedicated to observing two greenhouse gases, namely, $CO_2$ and $CH_4$, with two instruments, the Thermal And Near infrared Sensor for carbon Observation–Fourier Transform Spectrometer (TANSO-FTS) and the Cloud and Aerosol Imager (TANSO-CAI). The former has wide spectral coverage from VIS to thermal infrared (TIR) radiation, and the latter is a radiometer covering the ultraviolet, VIS and shortwave infrared (SWIR) spectral range to retrieve cloud and aerosol characteristics. The TANSO-FTS has three bands (bands 1, 2, and 3) at SWIR wavelengths centered at 760, 1600, and 2000 nm, respectively, and a band (band 4) at TIR wavelengths covering 5.56-14.3 $\mu$m.

GOSAT SIF data are retrieved using radiation spectral data at a retrieval window between 756.0 and 759.1 nm in the TANSO-FTS band 1 (Oshio et al., 2019). GOSAT SIF data are derived from the infilling of Fraunhofer lines retrieved from TANSO-FTS spectra minus the zero-level offset, which is an artifact signal resulting from nonlinearity in the analog circuit. The cloud pixel fraction (CPF) within the instantaneous field of view (IFOV) of the TANSO-FTS was computed using the integrated clear confidence level information in the cloud coverage data (TANSO-CAI Level 2 product). We used the CPF for cloud screening to remove the data contaminated by clouds and aerosols. The threshold for cloud screening was empirically set to CPF > 15%. Then, the observational geometries of the satellite observation zenith angle, solar zenith angle, and relative azimuth angle between GOSAT and the sun were computed for individual GOSAT observation points for angularity correction of the simulated SIF, as described in a later subsection.

### 2.2 Model description

#### 2.2.1 Process-based terrestrial ecosystem model: VISIT

This study used a process-based terrestrial ecosystem model, the Vegetation Integrative SImulator for Trace gases (VISIT; Ito, 2010, 2019), to simulate biophysical and biochemical processes. The VISIT is composed of multiple modules that represent matter flows within ecosystems and exchanges between the atmosphere and ecosystems. A box-flow scheme with eight carbon pools (leaf, stem, and root carbon for $C_3$ and $C_4$ plants, soil litter and humus) is adopted for the simulation of the carbon cycle in VISIT. GPP is represented as a function of the leaf area index (LAI), incident photosynthetically active radiation (PAR), air temperature and humidity, soil water content, and ambient $CO_2$ concentration. The absorption and diffusion of radiation and carbon assimilation are simulated using a two-component canopy model by de Pury and Farquhar (1997). Leaf photosynthetic capacity is regulated by leaf nitrogen concentration, and LAI is predicted by leaf carbon amount

and specific leaf area for each land cover type. This study classified global land cover into 16 land cover types based on the Terra and Aqua combined Moderate Resolution Imaging Spectroradiometer (MODIS) Land Cover Climate Modeling Grid (CMG) with International Geosphere-Biosphere Programme classification (MCD12C1-IGBP) (Sulla-Menashe and Friedl, 2018) with a spatial resolution of 0.3125 degrees.

## 2.2.2 SIF simulation

We simulated the spatiotemporal variability of SIF using a combination of the VISIT and SCOPE version 1.74 models. A diagram of the VISIT-SIF model system is shown in Fig. 1. The model system consists of biochemical/biophysical processes and geometric and radiative transfer processes. The former simulates the canopy structures and the radiative conditions within the canopy and the actual and potential electron transport rates for a given grid. The simulated electron transport rates are inputs for the quantum yield of chlorophyll fluorescence, and the absorbed photosynthetically active radiation (APAR) is used to calculate SIF. The latter simulates radiative transfer processes for the SIF emitted from the upper canopy. The practical operation manner to simplify the simulation of radiative transfer processes is given later in this subsection.

In the model system, the chlorophyll fluorescence on the top of vegetation canopy $F$ (mW m$^{-2}$ sr$^{-1}$ nm$^{-1}$) at the observation angle is described by a combination of the chlorophyll fluorescence emitted from sunlit and shaded leaves as follows:

$$F = F_{\text{sun}}\left(1 + r_{\text{shade/sun}}\right) \tag{1}$$

where $F_{\text{sun}}$ is the chlorophyll fluorescence emitted from sunlit leaves at the observation angle (mW m$^{-2}$ sr$^{-1}$ nm$^{-1}$) and $r_{\text{shade/sun}}$ is the ratio of chlorophyll fluorescence emitted from shaded leaves to $F_{\text{sun}}$, the details of which will be described later. $F_{\text{sun}}$ can be described by:

$$F_{\text{sun}} = \frac{\text{APAR}_{\text{sun}} \; \Phi_{\text{F,sun}} \; r_{\text{oz/sz}} \; f_{\text{u}}}{\pi} \tag{2}$$

where $\text{APAR}_{\text{sun}}$ and $\Phi_{\text{F,sun}}$ are APAR (W m$^{-2}$) and the quantum yield of chlorophyll fluorescence in sunlit leaves, respectively. $r_{\text{oz/sz}}$ is a correction factor for converting the chlorophyll fluorescence emitted from sunlit leaves to remote-sensed chlorophyll fluorescence in arbitrary observation directions. Here, APAR is the photosynthetically active radiation absorbed by canopy, which is the product of canopy-top irradiance, canopy reflectance, LAI and an attenuation coefficient that is a function of biome type and solar angle (Ito and Oikawa, 2002). $\text{APAR}_{\text{sun}}$ consists of the absorbed beam, diffuse, and scattered beam with the sunlit layer based on Farquhar model (de Pury and Farquhar 1997). Under the assumption that

VISIT simulates biogeochemical processes occurring within sunlit leaves where the viewing angle coincides with the sun zenith angle, we adopted APAR$_{sun}$ in Eq. (2). The variable $f_u$ represents the fraction of the SIF emitted in the upward direction to that in both the upward and downward directions at the canopy-level. It was obtained as the average fraction across the 60 canopy layers by operating the SCOPE model. The variable can be used to estimate canopy-level fluorescence by considering radiative transfer processes within the canopy layers; however, the reabsorption at single-leaf-level in fluorescence yield was not reflected. In addition, Equation (2) describes indirect incorporation of VISIT and SCOPE by multiplying $\Phi_{F,sun}$, $r_{oz/sz}$ and $f_u$. $\Phi_{F,sun}$ is calculated as a function of energy allocation between photochemistry and chlorophyll fluorescence at photosystem-level as follows:

$$\Phi_{F,sun} = \Phi_{Fm',sun} \left(1 - \Phi_{P,sun}\right) \tag{3}$$

where $\Phi_{Fm',sun}$ and $\Phi_{P,sun}$ are the quantum yield of fluorescence at light-acclimated leaves exposed to saturated irradiance and by actual photochemistry, respectively. $\Phi_{Fm',sun}$ is defined as the ratio of irradiance emitted as chlorophyll fluorescence to total irradiance as follows:

$$\Phi_{Fm',sun} = \frac{k_F}{k_F + k_D + k_{N,sun}} \tag{4}$$

where $k$ denotes the rate coefficient for chlorophyll fluorescence ($k_F$), for nonphotochemical quenching (NPQ) of dark-acclimated leaves ($k_D$), and for NPQ of sunlit leaves ($k_{N,sun}$). In this study, $k_F$ and $k_D$ are fixed at 0.05 and 0.95 according to Tol et al. (2014), respectively. $k_{N,sun}$ is represented using the following empirical equations (Flexas et al., 2002):

$$k_{N,sun} = (6.2473x - 0.5944)\,x \tag{5}$$

$$x = 1 - \frac{\Phi_{P,sun}}{\Phi_{P0}} \tag{6}$$

where x is the degree of light saturation and $\Phi_{P0}$ is the quantum yield for photochemistry in dark-acclimated leaves. $\Phi_{P0}$ is defined as follows:

$$\Phi_{P0} = \frac{k_P}{k_F + k_D + k_P} \tag{7}$$

where $k_P$ is the rate coefficient of irradiance emanated during photochemical reactions to the irradiance total ($k_P = 4.0$) (Tol et al., 2014).

$\Phi_{P,sun}$ is described as:

$$\Phi_{P,sun} = \Phi_{P0} \frac{J_{a,sun}}{J_{e,sun}} \tag{8}$$

where $J_{a,sun}$ and $J_{e,sun}$ ($\mu mol$ $m^{-2}$ $s^{-1}$) are the actual and potential electron transport rates, respectively. $J_{a,sun}$ is given by VISIT based on de Pury and Farquhar (1997) and van del Tol (2014) as:

$$J_{a,sun} = 4A_{sun} \frac{C_i + 2\Gamma^*}{C_i - \Gamma^*} \tag{9}$$

where $A_{sun}$ ($\mu mol$ $m^{-2}$ $s^{-1}$) is the $CO_2$ assimilation in sunlit leaves, $C_i$ (Pa) is the intercellular $CO_2$ partial pressure, and $\Gamma^*$ (Pa) is the $CO_2$ compensation point of photosynthesis. $A_{sun}$ is calculated as follows:

$$A_{sun} = \min (A_{c,sun}, A_{j,sun}) \tag{10}$$

$$A_{c,sun} = V_{cmax} \frac{C_i - \Gamma^*}{C_i + K'} \tag{11}$$

$$A_{j,sun} = J_{sun} \frac{C_i - \Gamma^*}{4(C_i + 2\Gamma^*)} \tag{12}$$

where $A_{c,sun}$ and $A_{j,sun}$ are the Rubisco- and RuBP regulation-limited $CO_2$ assimilation in sunlit leaves. $V_{cmax}$ ($\mu mol$ $m^{-2}$ $s^{-1}$) is the maximum carboxylation rate, which varies depending on LAI, canopy temperature, nitrogen condition and water stress.

$K'$ (Pa) is the effective Michaelis-Menten constant of Rubisco and is calculated based on land cover-specific parameters and temperature. $J_{sun}$ is the electron transport rate in sunlit leaves and is described as:

$$J_{sun} = \frac{I_{sun} + J_{max} - \sqrt{(I_{sun} + J_{max})^2 - 4\theta_l \, I_{sun} J_{max}}}{2\theta_l} \tag{13}$$

where $I_{sun}$ ($\mu mol$ $m^{-2}$ $s^{-1}$) is the absorbed photosynthetic photon flux density, $J_{max}$ is the maximum electron transport rate ($\mu mol$ $m^{-2}$ $s^{-1}$), $\theta_l$ is the curvature showing the leaf response to irradiance for electron transport (=0.7; de Pury and Farquhar, 1997). $J_{max}$ is calculated as a function of $V_{cmax}$ along with land cover-specific parameters that vary with canopy temperature and nitrogen conditions in VISIT. $J_{e,sun}$ is calculated as follows:

$$J_{e,sun} = I_{sun} \cdot \Phi_{P0} \tag{14}$$

The geometric coefficients of $r_{oz/sz}$ and $r_{shade/sun}$ constantly vary in space and time with changes in the solar zenith angle (SZ), observation zenith angle (OZ), and relative azimuth angle of the solar and observation directions (AZ). We employed a low-computational-cost approach using a look-up table (LUT) to formulate the geometric coefficients for every observation point instead of solving the radiative transfer process directly with RTM. The probability distributions of $r_{oz/sz}$ and $r_{shade/sun}$ were computed by shifting the angle parameters SZ, OZ and AZ in 10°, 10° and 30° angle steps, respectively, and the values of LAI and solar radiation (SRAD; W m$^{-2}$) in 0.5 and 200 W m$^{-2}$ steps using the SCOPE model with the parameters set in Table A1, respectively, as follows:

$$r_{oz/sz} = \frac{F_{SCOPE,sun}(LAI, SZ, OZ, AZ, SRAD)}{F_{SCOPE,sun,sz=oz}(LAI, SZ, OZ', AZ', SRAD)} \tag{15}$$

$$r_{shade/sun} = \frac{F_{SCOPE,shade}(LAI, SZ, OZ, AZ, SRAD)}{F_{SCOPE,sun}(LAI, SZ, OZ, AZ, SRAD)} \tag{16}$$

where $F_{SCOPE,sun}$, $F_{SCOPE,sun,sz=oz}$, and $F_{SCOPE,shade}$ are the chlorophyll fluorescence from sunlit leaves when the observation direction and solar incoming direction are located along one axis (OZ' = SZ, AZ' = 0) and from shaded leaves, all of which were computed using the SCOPE model. When implementing Eqs. (15) and (16) in the simulation of SIF variability, LAI and SRAD are driven by VISIT, and the angle parameters are computed from the geometric information obtained for every GOSAT observation. A brief overview of parameter selection for Eqs. (15) and (16) is provided in Appendix B.

The simulated SIF $F$ given by Eq. (1) is ideally denoted as the total fluorescence emission that occurs in the full chlorophyll emission spectrum. To compare $F$ with GOSAT SIF retrievals, radiance should be converted to chlorophyll emission at wavelengths between 756.0 and 759.1 nm (SIF$_{756}$). We computed the SIF$_{756}$ by multiplying the value by a correction factor as follows:

$$SIF_{756} = F \cdot r_{756} \tag{17}$$

where $r_{756}$ is the factor with respect to the fraction of SIF emission at wavelengths between 756.0 and 759.1 nm to the total chlorophyll emission ranging between 641 and 850 nm. The shape of the SIF emission spectra varies with the reabsorption process and depends on the leaf chlorophyll content. This study used a simple approach to describe $r_{756}$ as a function of the contents of chlorophyll a, chlorophyll b and carotenoids per unit of leaf area ($C_{ab}$; μg cm$^{-1}$) (Fig. C1). This study used the land cover-specific values of $C_{ab}$ from the study of Norton et al. (2019). Here, the relationship between $C_{ab}$ and $r_{756}$ was simulated by operating the SCOPE model with changes in the LAI but with less sensitivity of $r_{756}$ to changes in LAI values

between 1 and 8. We used a regression formula ($r_{756} = 1.2 \cdot 10^{-3} \ln (C_{ab}) + 4.7 \cdot 10^{-3}$) to estimate $r_{756}$ for whole land cover types regardless of changes in the LAI (Table A2 and Fig. C1).

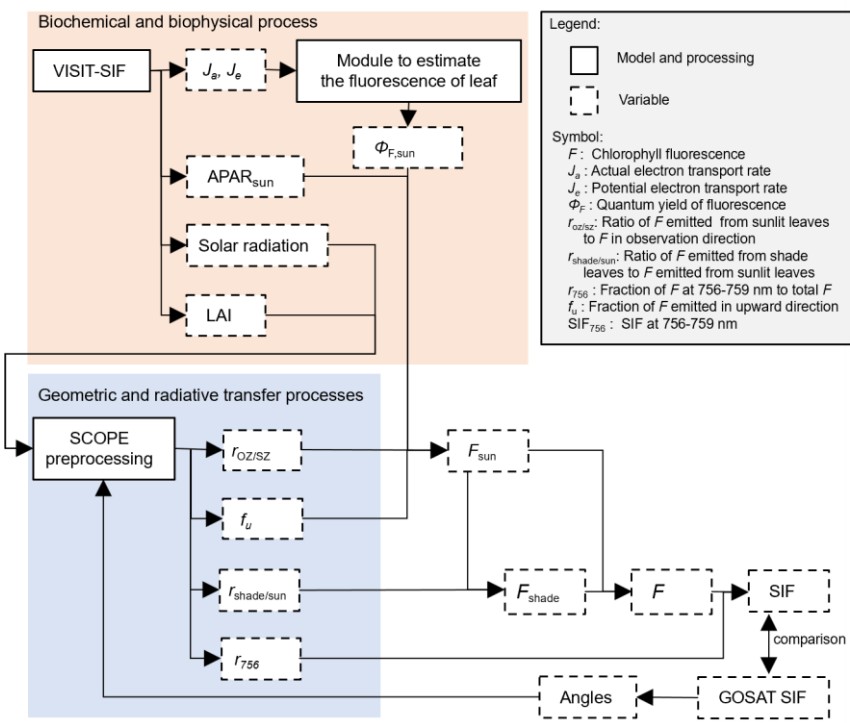

**Figure 1.** Schematic diagram of the VISIT-SIF model system.

## 2.3 Model processing

Initialization of the VISIT model was achieved by a spin-up run of 900 years with repeated use of climate forces. Then, the model system was operated over 7 years for 2009-2015 in hourly time steps. In the following analyses, we used only the
250 model results that were simulated at 13:00 local time and with a cloud fraction lower than 0.5. The composite data of the European Centre for Medium Range Weather Forecasts (ECMWF) Reanalysis-interim (ERA-interim) at a 3-hour resolution (Dee et al., 2011) and the National Centers for Environmental Prediction (NCEP) Climate Forecast System Reanalysis (CFSR) at an hourly resolution (Saha et al., 2010) were used as the climate forcings. Once all the meteorological variables in the CFSR were scaled to those of the reference datasets on a month-to-month basis, the Climate Research Unit (CRU) Time-
255 Series (TS) (Harris et al., 2019) for precipitation and specific humidity and the ERA interim for other variables were used. Regarding the specific humidity, the ratio of the vapor pressure in the CRU TS to that in the CFSR, which was computed from the specific humidity, was substituted to scale the CFSR values. Then, the hourly composite data were constructed by

adding hourly meteorological fluctuations derived from the CFSR to the 3-hourly ERA interim datasets, for which the specific humidity was calculated using the dew point and surface pressure data. Deviations between the hourly CFSR variables and 3-hourly means were used to determine the hourly fluctuations. For the wind velocities, the CFSR data were used without corrections. In this process, the ERA interim and CRU TS datasets were interpolated onto the T382 CFSR grid.

## 3 Results

### 3.1 Global VISIT-SIF simulation and comparison with GOSAT SIF

We present the global distribution of the mean SIF from the GOSAT retrievals and model simulations for 2009 and 2015 at a spatial resolution of 2.5 degrees (Fig. 2). The map of the model simulations was generated by using only the data corresponding to the locations and times of available GOSAT retrievals. GOSAT retrievals show a pronounced increase in SIF in the tropics in the Amazon, Borneo and New Guinea, with an approximate value of 0.67 mW $m^{-2}$ $sr^{-1}$ $nm^{-1}$. The intensities of SIF showed a gradual decrease with increasing latitude, whereas large variations in SIF are shown with increasing longitude. In boreal forests, the intensities of SIF for satellite observations, approximately 0.21 mW $m^{-2}$ $sr^{-1}$ $nm^{-1}$, are lower than those in the mid-latitudinal zone, with approximately 0.35 mW $m^{-2}$ $sr^{-1}$ $nm^{-1}$. The model simulations showed spatial patterns similar to those of satellite observations worldwide. The global mean value and standard deviation of the SIF for the model simulations are $0.51 \pm 0.39$ mW $m^{-2}$ $sr^{-1}$ $nm^{-1}$, which are in good agreement with the satellite observations, with a value of $0.46 \pm 0.42$ mW $m^{-2}$ $sr^{-1}$ $nm^{-1}$. However, at the regional scale, differences in SIF between model simulations and satellite observations are identified, including overestimation in Southeast Africa and western North America and underestimation in central Amazon.

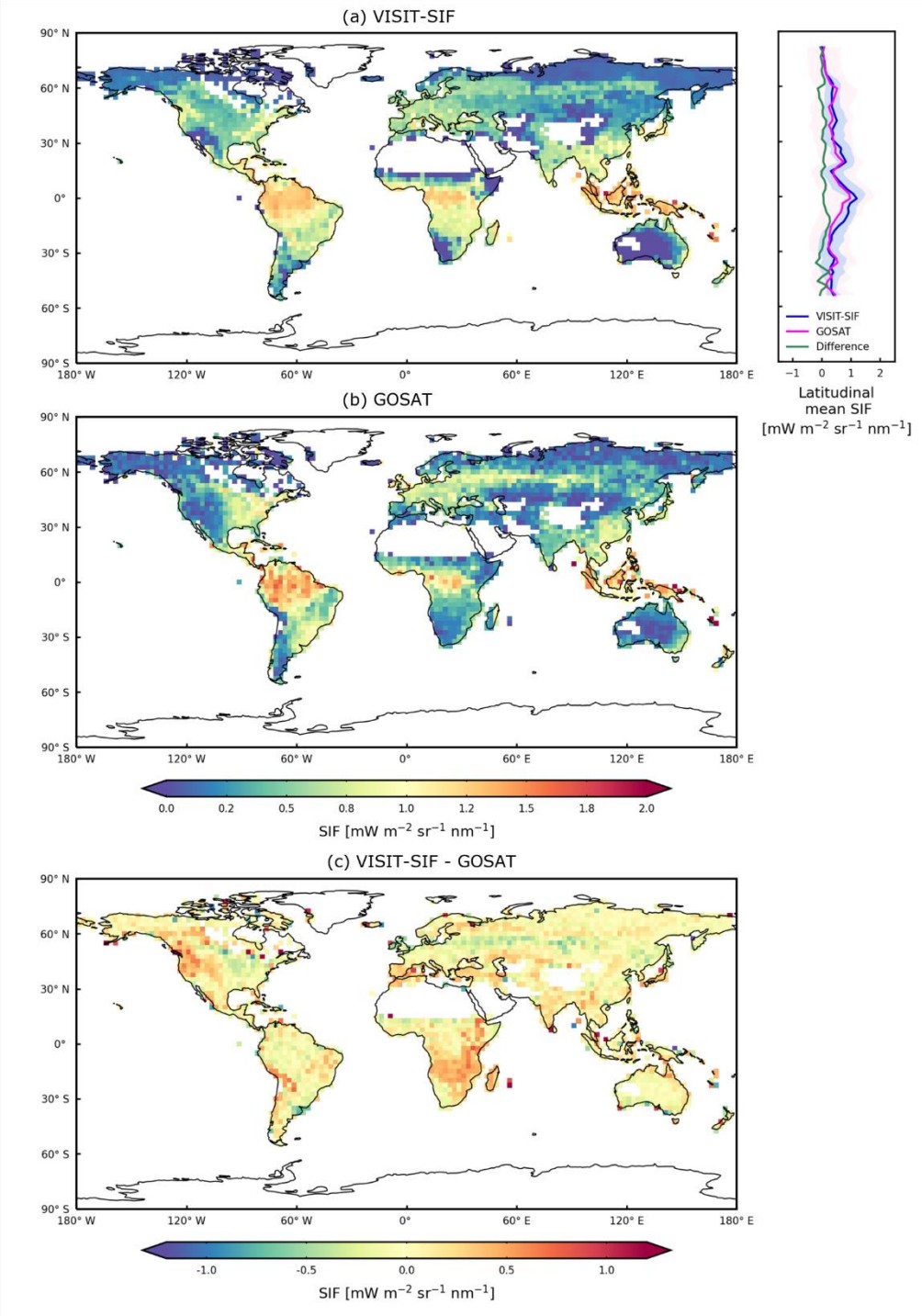

**Figure 2.** Global distributions of annual mean SIF (mW m$^{-2}$ sr$^{-1}$ nm$^{-1}$) for (a) VISIT-SIF simulations, (b) GOSAT SIF retrievals, and (c) their differences for 2009 and 2015. The spatial resolution is aggregated to a 2.5-degree grid.

A direct comparison of the mean SIF between the satellite observations and model simulations (Fig. 2) is shown in Fig. 3a. According to this comparison, the two data points are correlated (correlation coefficient R = 0.76; root mean squared error RMSE = 0.29 mW m$^{-2}$ sr$^{-1}$ nm$^{-1}$) and follow the 1:1 line, indicating similar intensities. This suggested that VISIT-SIF can produce proper spatial variability in GOSAT SIF retrievals, whereas deviations from the 1:1 line and outliers are identified between the two datasets, depending on the region, as shown in Fig. 2c. These differences may be due to various factors: the random retrieval error of GOSAT SIF, which is approximately 0.2 mW m$^{-2}$ sr$^{-1}$ nm$^{-1}$ (Oshio et al., 2019);

variations in SIF across space used for spatial aggregation; and insufficient parameterization of SIF variability at the local scale in the model.

As described in section 2.2.2, this study simulated GOSAT SIF retrievals by accounting for the observational geometry using the parameters $r_{oz/sz}$ and $r_{shade/sun}$. The performances of these geometrical correction parameters in the

simulations are shown in Fig. 3b, which indicates that the SIF was simulated without geometrical correction. Here, the SIF variability was simulated by replacing $F$ in Eq. (17) with $F_{sun}$ in Eq. (2), where $r_{oz/sz} = 1$. The SIF values are obviously greater than those of the satellite observations, and the differences are greater (RMSE = 0.50 mW m$^{-2}$ sr$^{-1}$ nm$^{-1}$) than the differences in the simulated SIF with geometric correction, as shown in Fig. 3a. Variability of SIF values to the geometrical correlation parameters is demonstrated in Fig. B1, which compares SIF values simulated by shifting the angle parameters

and biophysical parameters in arbitrary step, respectively. There are substantial changes in SIF values for the angle parameters, SZ and OZ. This is partly expected because GOSAT has a two-axis pointing mechanism with pointing angles of ±35 degrees and ±20 degrees in the cross-track (CT) and along-track (AT) directions, respectively, and points at any target observation area mainly by rotating in the CT direction (Kuze et al. 2012). The geometric relationships among the incidence angle of the emission to the sensor, solar azimuth, and orientation of leaves can vary widely between observations, even for

adjacent scans. Accordingly, the differences without geometrical correction shown in Fig. 3 suggest that the observational geometry is critical information for obtaining more reliable simulations of GOSAT SIF retrievals. As an additional validation, we have verified the differences between using different observation angles and a fixed observation angle, which are shown in Appendix D.

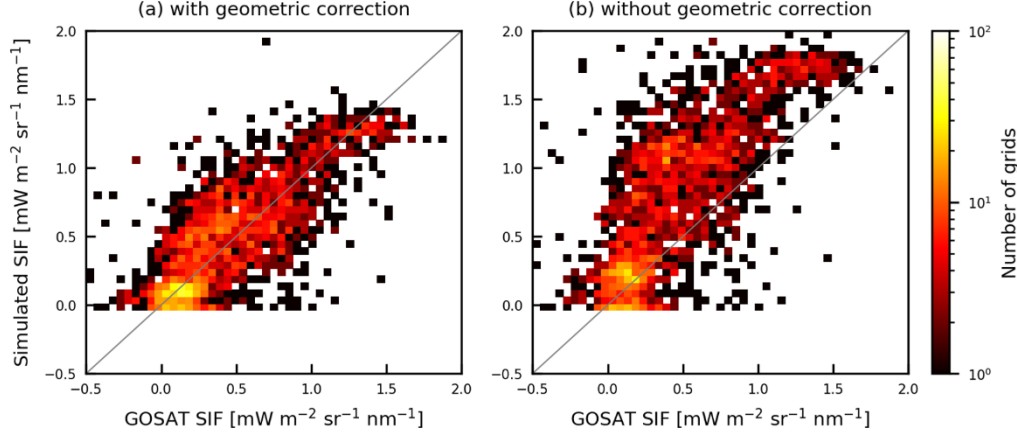


**Figure 3.** Relationship between the GOSAT SIF retrievals (x axis) and simulated SIF (y axis) (a) with geometric correction using the parameters $r_{oz/sz}$ and $r_{shade/sun}$ and (b) without correction. The scatterplots represent the mean annual values aggregated to a 2.5-degree grid. The color bar shows the number of data points.

To evaluate the simulated SIF intensity across land cover types, the mean SIF values for 2009 and 2015 at a 2.5 degree spatial resolution obtained from satellite observations and model simulations were compared for each land cover type. Fig. 4 shows box plots of the mean SIF values for 12 land cover types. Overall, the satellite observations showed wide dispersion along with negative SIF values regardless of the land cover type. The negative SIF values are not actual physical quantities because of the presence of retrieval noise, but this study used all the satellite observations without discarding the 315    negative values to prevent significant biases in the probability distribution of the SIF variability. We found that model simulations exhibited land cover-specific variation consistent with that of satellite observations: higher SIF values for evergreen broadleaf forests, with mean values of 0.99 and 0.96 mW m$^{-2}$ sr$^{-1}$ nm$^{-1}$ for model simulations and satellite observations, respectively, and lower SIF values for open shrublands and grasslands, with mean values of 0.17 and 0.31 mW m$^{-2}$ sr$^{-1}$ nm$^{-1}$ for model simulations and 0.11 and 0.16 mW m$^{-2}$ sr$^{-1}$ nm$^{-1}$ for satellite observations, respectively. However, the 320    divergence in the mean values between the model simulations and satellite observations increased for some land cover types, especially for deciduous forest types: deciduous needleleaf forests (0.36 mW m$^{-2}$ sr$^{-1}$ nm$^{-1}$) and deciduous broadleaf forests (0.46 mW m$^{-2}$ sr$^{-1}$ nm$^{-1}$). This suggested that there is some inconsistency in the seasonal cycle of the simulated and observed SIF variations for these land cover types. A detailed analysis of the seasonal variability is given in the following subsection.

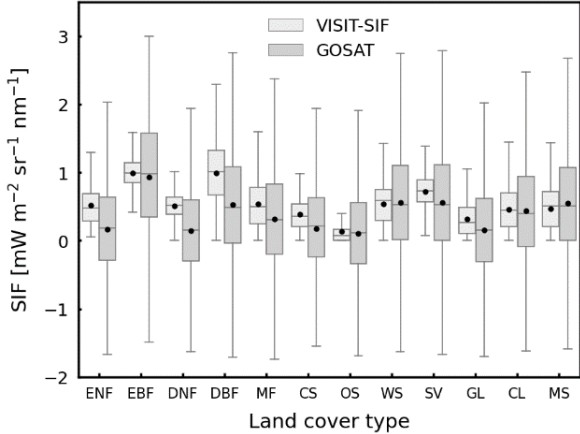


**Figure 4.** Box plots of annual mean values (2009-2015) of VISIT-SIF simulations (light gray) and GOSAT retrievals (gray) on a 2.5-degree grid for 12 land cover types: ENF: evergreen needleleaf forests; EBF: evergreen broadleaf forests; DNF: deciduous needleleaf forests; DBF: deciduous broadleaf forests; MF: mixed forests; CS: closed shrublands; OS: open shrublands; WS: woody savannas; SV: savannas; GL: grasslands; CL: cropland; and MS: mosaic. The black dots represent
the mean values.

## 3.2 Seasonal SIF variability

To compare the seasonal variability in the simulated SIF with that of satellite observations, the global terrestrial area was divided into 42 subcontinental regions based on the source regions for global $CO_2$ and $CH_4$ source and sink estimates that
have been applied in the GOSAT Level 4 data product. The boundaries of these source regions are shown in Fig. E1. Fig. 5 shows the seasonal variability in the monthly mean SIF averaged over 7 years (2009-2015) for the model simulations and satellite observations and their differences over the 42 regions. The seasonal cycles appear rather similar for model simulations and satellite observations, with relatively large amplitudes in the mid-latitude regions and small amplitudes in the tropics and high-latitude regions. Seasonal variations in the model simulations vary smoothly relative to those based on
the satellite observations in the regions where GOSAT retrievals showed large fluctuations with time; these include region 8, which is dominated by temperate deciduous forests; region 14, temperate grasslands and shrublands; region 22, grasslands and savannas; region 17, tropical forests and savannas; and region 16, tropical forests, savannas, and deserts. The variations in these regions were 0.82, 0.52, 0.48, 0.16, and 0.37 mW $m^{-2}$ $sr^{-1}$ $nm^{-1}$ for the model simulations and 1.15, 0.82, 0.72, 0.67, and 0.65 mW $m^{-2}$ $sr^{-1}$ $nm^{-1}$ for the satellite observations, respectively. For the maximum differences in the monthly mean
values between the model simulations and satellite observations, the model overestimates the intensity of SIF by up to 0.68, 0.66, 0.64, 0.55, and 0.52 mW $m^{-2}$ $sr^{-1}$ $nm^{-1}$ in region 22, region 24 (dominated by savannas), region 30 (tropical forests and deserts), region 29 (deserts), and region 15 (grasslands and savannas), respectively. These overestimates in the model

resulted from inconsistencies in vegetation phenology during the dormant season: the model estimates vigorous photosynthetic activities, while satellite observations depict attenuation of photosynthesis.


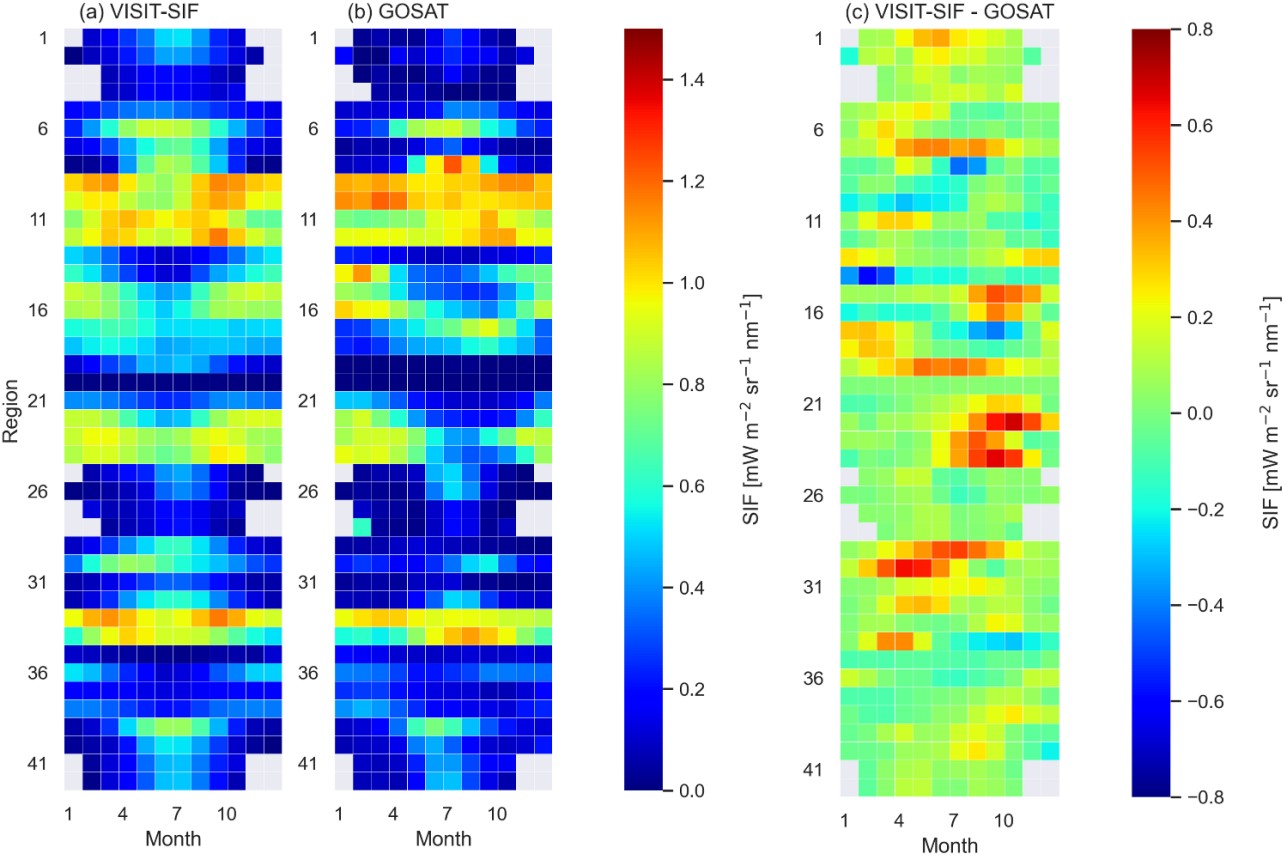

**Figure 5.** The seasonal variability in SIF averaged for 2009 and 2015 for (a) VISIT-SIF simulations and (b) GOSAT retrievals in 42 subcontinental regions and their differences and (c) the difference between VISIT-SIF simulation and GOSAT retrievals. Gray grid cells indicate that no data were available.


To quantify the differences in the seasonal variations and intensities between the model simulations and satellite observations, the RMSE and $R^2$ were computed for each region based on the monthly mean SIF values (Fig. 6). Strong linear relationships and lower RMSE values were observed over the subarctic zone on the Eurasian continent (regions 25, 26, 41, and 42), with $R^2 > 0.88$ and RMSE $< 0.08$ mW m$^{-2}$ sr$^{-1}$ nm$^{-1}$. In contrast, weaker relationships were found in southeastern Africa (regions 22 and 24) and the Indian subcontinent (region 30), with $R^2 < 0.03$ and RMSE $> 0.33$ mW m$^{-2}$ sr$^{-1}$ nm$^{-1}$. We found that these discrepancies occurred for the late dry season to early rainy season when the number of valid retrievals was not much lower than that in the rainy season. This relatively high level of data acquisition can reduce random retrieval

errors; thus, the large differences in the seasonal variations in SIF and its intensity in the 3 regions suggested that the model representation could be poorly constrained, especially for the vegetation response to water stress over arid and semiarid regions, perhaps due to a lack of ground-based observations.

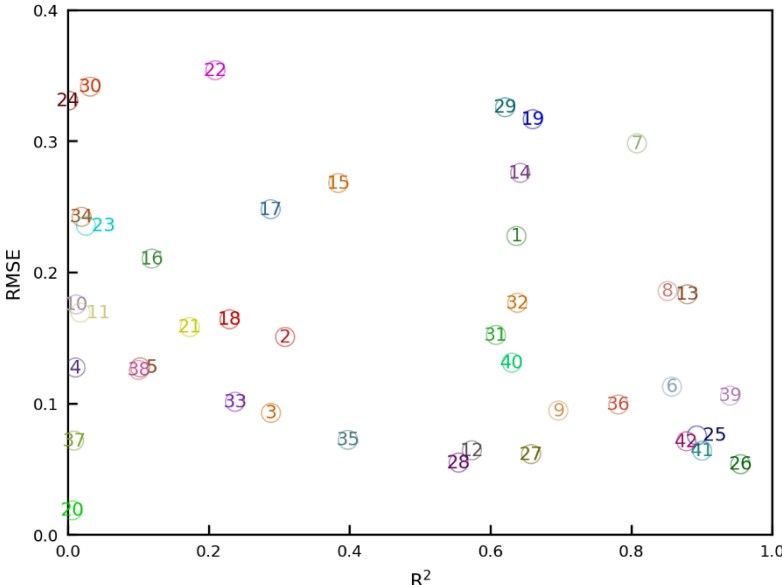

**Figure 6.** The relationships between $R^2$ and RMSE for the mean monthly variability in the VISIT-SIF simulations and GOSAT retrievals in the 42 subcontinental regions. The numbers in the figure correspond to the regional IDs shown in Fig. E1.

We further examined monthly mean SIF variations from 2009 to 2015 for 9 selected regions extending from the tropics to the mid-latitudinal region (Fig. 7). The model simulations appear to capture the seasonal cycles of satellite observations except for region 23 with tropical forests, grasslands, and savannas and region 34 with tropical forests. In particular, for region 23, the most striking difference was observed for July and September, when satellite observations showed a distinct decrease, while model simulations indicated weaker seasonal variability. In the tropical forest area in this region, the seasonal precipitation cycle has weakened, with a significant increase in the boreal winter dry season and a decrease in the boreal spring wet season, which may be driven by changes in sea surface temperature, particularly in the Atlantic and Indian Oceans (Wang et al., 2021). However, the GOSAT SIF yields a distinct seasonality. Note that region 23 has fewer valid retrievals due to the existence of continuous clouds. The spatiotemporal variations in SIF variability, as well as regional meteorological and hydrological cycles, in tropical regions need further investigation.

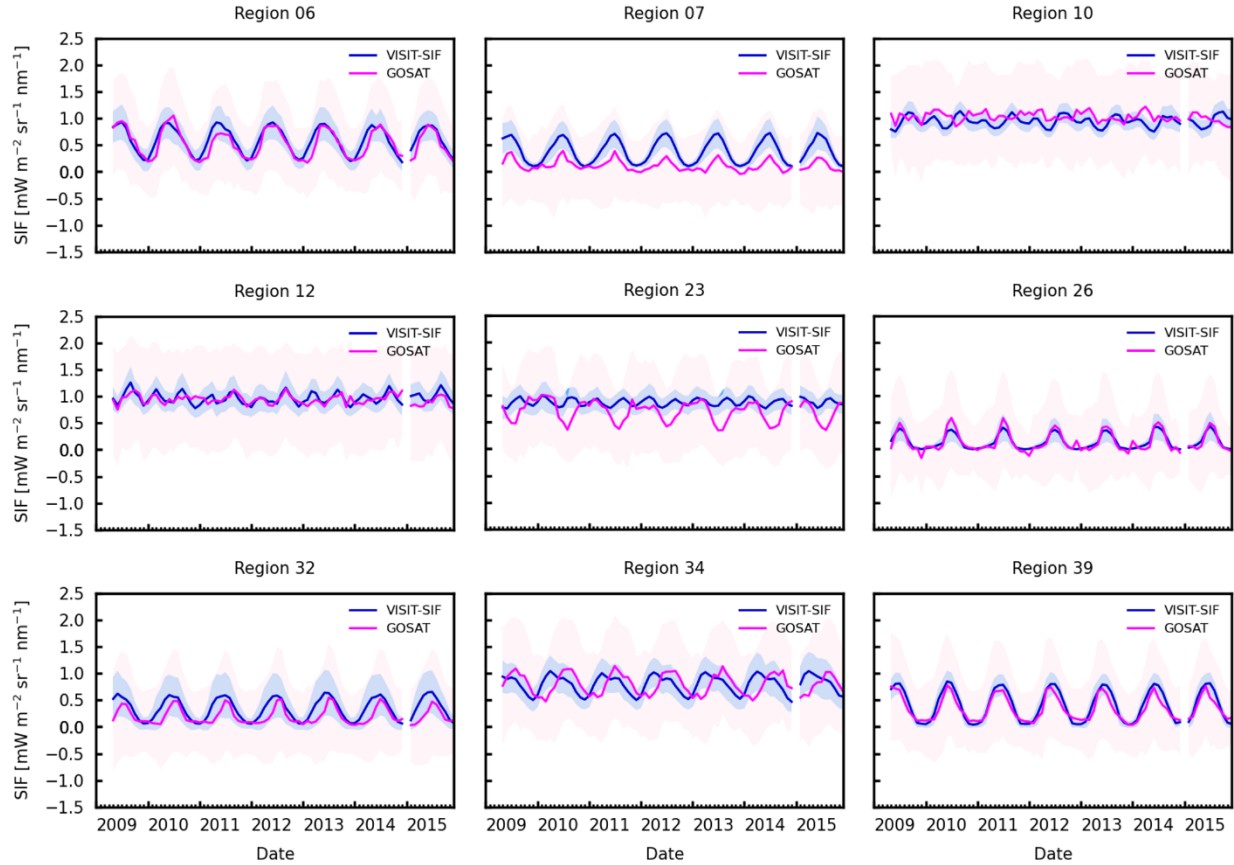

**Figure 7.** Time series of monthly SIF for VISIT-SIF simulations (blue) and GOSAT retrievals (pink) for 2009 and 2015 in 9 selected regions: 6, 7, 10, 12, 23, 26, 32, 34 and 39. The shaded areas shows the standard deviations.

## 4. Discussion

Interest in satellite-based SIF observations has grown since successful global SIF retrievals have been achieved with GOSAT, with the hope that these observations can stimulate our understanding of terrestrial ecosystem dynamics. However, as SIF is only a small amount of energy being reemitted in concert with photochemical reactions and heat dissipation, a biochemical process model is needed to connect SIF retrievals with ecosystem-level processes. Various underlying models are still in development (e.g., Parazoo et al., 2020), and the retrieved SIF intensity significantly varies among satellites with different observed spectral ranges, observational times, and angles between the viewing and sun directions (Oshio et al., 2019; Murakami et al., 2024). Here, we developed a VISIT-SIF biogeochemical process model to estimate GOSAT SIF retrievals

by incorporating observational geometry using the parameters $r_{oz/sz}$ and $r_{shade/sun}$. This geometric correction is necessary for predicting the GOSAT SIF retrievals precisely, as shown in Figs. 3 and C1, partly because GOSAT has a wide range of observation angles due to the operation of the two-axis pointing mechanism for precise viewing of target locations (Kuze et al., 2012).

    By using SCOPE in a simplified manner, the radiative transfer process between canopy layers to calculate SIF in
arbitrary observation direction can be computed; however, uncertainty remains in the calculation of chlorophyll fluorescence yield at leaf- and canopy-level. While the reabsorption by chlorophyll was considered in the calculation of $r_{756}$, the internal reabsorption at the single-leaf-level was not directly considered in the fluorescence yield in Eq. (2). Li et al. (2022) used the simplified scattering fluorescence coefficient in SCOPE, which estimates the chlorophyll fluorescence from the excitation light, to convert the fluorescence yield at the photosystem level to the leaf level. However, since VISIT-SIF and SCOPE
differ in the model structure of canopy layers and spectral calculations, a conversion approach for fluorescence yield at leaf level has not been implemented in VISIT-SIF. The conversion of fluorescence yield from photosystem level to leaf-level is a challenge for future improvements of VISIT-SIF.

    The new model system presented here still has some room for improvement; however, it is appropriate for estimating the global distribution with respect to the mean values of the GOSAT SIF retrievals (Fig. 2). The comparison with
GOSAT SIF retrievals provided insights into how SIF emitted from the terrestrial biosphere responds to seasonal changes in meteorological and hydrological conditions in a given region. This comparison revealed that the seasonal variability in the simulated SIF indicated an insufficient decline for the late dry season in arid and semiarid regions relative to that of satellite observations (Figs. 5 and 7). Similarly, Parazoo et al. (2020) reported that insufficient NPQ formulation under drought conditions, especially for lower-latitude regions, can result in a weak decrease in SIF with little to no sensitivity to water
stress. As shown in section 2.2.2, this study calculates NPQ (= $k_{N, sun}$) using the variables $V_{cmax}$, $J_{max}$ and $I_{sun}$, which vary in response to canopy structure and environmental stresses in the model, such as leaf area index, temperature, and water and light limitations. Accordingly, our simulations of GOSAT SIF retrievals using an initial configuration of ecophysiological model parameters demonstrated that careful improvements in model representation are necessary for estimating NPQ dynamics and related biophysical processes, particularly as they relate to water stress in arid and semiarid regions.

In terms of the NPQ response to water stress, soil water content is a crucial factor that directly restricts $V_{cmax}$ and thus $J_{max}$ in the VISIT model, as well as temperature and intercellular $CO_2$ concentration (Ito and Oikawa, 2002). Water stress is expressed as an empirical function of the soil water content, with coefficients for the field capacity of soil water, soil moisture photosynthesis limitation, and weight factors of water stress sensitivity, all of which have been validated using field observational data at 17 sites worldwide. The apparent discrepancy in the simulated SIF seasonality in arid and semiarid
regions may be primarily attributed to the poor representation of water stress using the empirical relationship and the limited amount of available validation data. Indeed, despite the obvious importance of water stress, the physiological mechanisms

underlying the relationship between photosynthesis and water stress have not been well characterized, and a more mechanistic understanding is needed.

For parameter optimization using satellite SIF retrievals, Norton et al. (2018) proposed a data assimilation
framework to minimize model-observation misfitting by constraining uncertainty in some key parameters, such as $V_{cmax}$ and $C_{ab}$, using satellite SIF retrievals as assimilated observations. As these parameters directly or indirectly define the photosynthetic rate, the posterior parameters demonstrated a successful reduction in uncertainty in global GPP estimates. Their results encouraged us to use a data assimilation framework to combine GOSAT SIF retrievals and VISIT-SIF, which may provide the benefit of constraining SIF and improving GPP estimates. Saito et al. (2014) constrained VISIT parameters
by incorporating atmospheric $CO_2$ concentration observations in a data assimilation system, but GOSAT SIF retrievals have not yet been tested to optimize VISIT-SIF parameters. Thus, optimizing VISIT-SIF parameters would be our next step for improving model representations of SIF variability and biophysical processes on a global scale, as well as further improving model formulations associated with SIF variability.

Satellite SIF observations provided us with a new indicator of photosynthetic capacity on a global scale. Available
satellite sensors capable of SIF retrievals include the Global Ozone Monitoring Experiment-2 (GOME-2) aboard the Meteorological Operational Satellite Program of Europe (MetOp) (Joiner et al., 2011), the Orbiting Carbon Observatory-2 (OCO-2) (Sun et al., 2017), and the TROPOsheric Monitoring Instrument (TROPOMI) abord the Sentinel-5 Precursor (S5p) (Zhang et al., 2019), as well as TANSO-FTS aboard GOSAT and TANSO-FTS-2 aboard GOSAT-2, which is the successor of GOSAT (Mohammed et al., 2019). Although these satellite sensors are designed for atmospheric studies and are not
dedicated to SIF monitoring, these SIF retrievals allow for the investigation of ecosystem responses to environmental stresses even at the local scale (e.g., Lee et al., 2013; Murakami et al., 2024). This study utilized GOSAT SIF retrievals to evaluate the newly developed VISIT-SIF model, which demonstrated the ability to express seasonal SIF variability even in areas lacking ground-surface observations. The measurement coverage is not always sufficient in the tropics, which are often covered by clouds. By utilizing other satellite SIF retrievals that were observed with different spectral ranges, IFOV,
measurement coverage, and on-orbit operation will complement each other for tracking variations in SIF and GPP with high accuracy and high spatial and temporal resolutions.

## 5. Conclusions

We developed a new biochemical process model to simulate GOSAT SIF retrievals. The SIF variability emitted at the top of
the canopy is expressed as a combination of the chlorophyll fluorescence emitted from sunlit and shaded leaves as determined by the SCOPE model. The model was operated with an hourly time step and a spatial resolution of 0.3125 degrees for 2009 and 2015, and a geometrical correction was included to account for changes in the SIF intensity depending

on the viewing angle of the sensor and the direction of the sun. Then, the simulated SIFs were compared with the GOSAT SIFs using only the data corresponding to the location and time of the valid GOSAT observations. An important first step

was to evaluate the global distribution of mean SIF values. The comparison of the model simulations with the GOSAT SIF retrievals showed consistency overall, with global mean values of $0.51 \pm 0.39$ and $0.46 \pm 0.42$ mW m$^{-2}$ sr$^{-1}$ nm$^{-1}$ for the model simulations and satellite observations, respectively, with an RMSE = 0.29 mW m$^{-2}$ sr$^{-1}$ nm$^{-1}$. We also compared the seasonal variability in SIF over the 42 subcontinental regions. This comparison indicated overestimates of simulated SIF during the dormant season in arid and semiarid regions, with less sensitivity to water stress. This study is still only a first step toward a

comprehensive understanding of global SIF variability and its interaction with biophysical processes.

## Appendix A: Input parameters

**Table A1.** Parameters of SCOPE used for computing and validating $r_{oz/sz}$ and $r_{shade/sun}$.

| Parameter | Symbol | Unit | Value or range |
|---|---|---|---|
| Solar zenith angle | SZ | degree | 0-75 |
| Observing zenith angle | OZ | degree | 0-75 |
| Relative azimuth | AZ | degree | 0-180 |
| Leaf area index | LAI | m2 m-2 | 0-10 |
| Incoming short wave radiation | SRAD | W m$^{-2}$ | 0-1000 |
| Air temperature | $T_a$ | °C | 20 |
| Air puressure | $p$ | hPa | 970 |
| Atmospheric vapour pressure | ea | hPa | 15 |
| Roughness length for momentum of the canopy | zo | m | 0.25 |
| Displacement height | $d$ | m | 1.34 |
| Leaf boundary resistance | rb | s m$^{-1}$ | 10 |
| Leaf angle distribution | LID (LIDF$_a$, LIDF$_b$) | | Spherical |
| Leaf angle distribution parameter a | LIDF$_a$ | | -0.35 |
| Leaf angle distribution parameter b | LIDF$_b$ | | -0.15 |
| Canopy height | $H_c$ | m | 2 |
| Leaf width | $w$ | m | 0.1 |
| Maximum carboxylation rate | $V_{cmax}$ | µmol m$^{-2}$ s$^{-1}$ | 60 |
| Chlorophyll a+b content | $C_{ab}$ | µg cm$^{-2}$ | 40 |
| Carotenoid content | $C_{ca}$ | µg cm$^{-2}$ | |
| Dry matter content | $C_{dm}$ | g cm | 0.012 |
| Leaf equivalent water thickness | $C_w$ | cm | 0.009 |
| Leaf thermal reflectance | $\rho$ (thermal) | | 0.01 |
| Leaf thermal transmittance | $\tau$ (thermal) | | 0.01 |

**Table A2.** The values of $r_{756}$ for each land cover type as estimated using the regression formula shown in Fig. C1.

| Land cover type | $r_{756}$ |
| --- | --- |
| Evergreen needleleaf forest | 0.0087 |
| Evergreen broadleaf forest | 0.0087 |
| Deciduous needleleaf forest | 0.0082 |
| Deciduous broadleaf forest | 0.0091 |
| Mixed forest | 0.0085 |
| Closed shrublands | 0.0073 |
| Open shrublands | 0.0082 |
| Woody savannas | 0.0085 |
| Savannas | 0.0085 |
| Grasslands | 0.0082 |
| Permanent wetlands | 0.0076 |
| Croplands | 0.0089 |
| Urban and developed area | 0.0080 |
| Cropland/natural vegetation mosaic | 0.0080 |
| Snow and ice | 0.0080 |
| Barren or sparsely vegetated | 0.0080 |

**Appendix B: Evaluation of $r_{oz/sz}$ and $r_{sd}$ in LUT and SCOPE**

As a preliminary validation for the input parameters in SCOPE, the impact of $r_{oz/sz}$ and $r_{sd}$ on $F$ in Eq. (1) when changing each input parameter in Table A1 was examined (Fig. B1). The impact on $F$ was estimated using SCOPE as the difference of $r_{oz/sz}$ and $1+r_{sd}$ in Eq. (1) and (2) compared to reference values ($r_{oz/sz}=0.5$ and $1+r_{sd}=1.12$). $r_{oz/sz}$ increases when SZ and OZ are close to each other, approaching a value of 1. As an exception, $r_{oz/sz}$ can exceed 1 when SZ is close to nadir, the OZ is lower value than SZ, and the LAI is lower than 2. The value of $r_{sd}$ is $0.11\pm0.05$, and it becomes 0 when OZ and SZ are identical. The reference values were determined from LUT by fixing OZ to 0 (nadir) and setting the parameters (LAI = 5, SRAD = 800 w m$^{-2}$, SZ = 30°) close to the average values in region 39. Since OZ, SZ and LAI have a significant impact on SIF calculation, they were selected as input parameters for Eqs. (15) and (16). Under the nadir-based condition, AZ had no influence on $F$. However, the angle between the solar incident direction and the observation direction, determined by SZ, OZ and AZ have a complex impact on $r_{oz/sz}$ and $r_{sd}$, selecting AZ as an input parameter. SRAD is a fundamental input variable for radiative transfer calculations and was applied in the LUT to align the light environment with VISIT. Variables such as $w$, $H_c$, and $C_{ab}$ have a relatively large effect on SIF calculations; however, since they are not computed within VISIT, fixed values were applied to them. SRAD is a fundamental input variable for radiative transfer calculations and was applied in the LUT to align the light environment with VISIT, although it caused only a small change in $r_{oz/sz}$ and $r_{sd}$. The variables such as $w$, $H_c$, and $C_{ab}$ have a relatively large impact on SIF calculations, however, since they are not computed within VISIT, fixed values were applied to them. LID also has a significant impact, but since there is no reasonable method for setting LID in global calculations for VISIT, it was fixed to spherical. The impact of other parameters fell within $\pm3\%$ and they were not selected as input parameters.

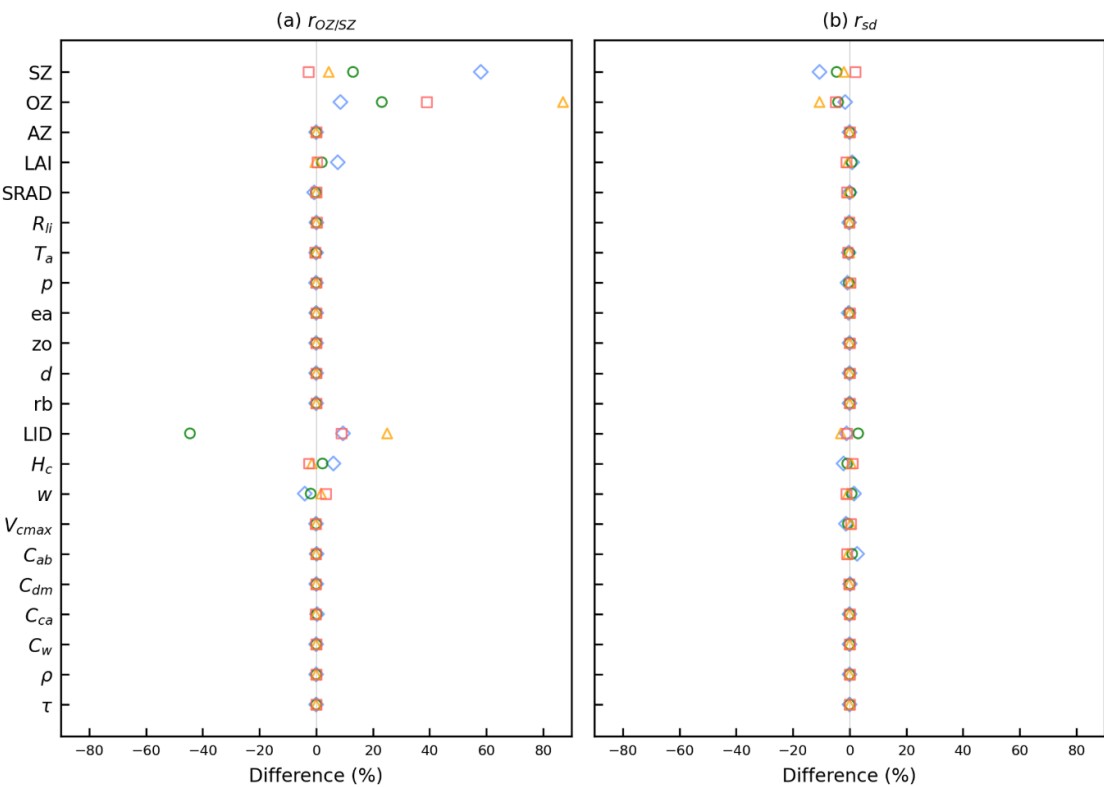

| Parameter | Reference value | ◇ | ○ | △ | □ |
|---|---|---|---|---|---|
| SZ | 30° | 0° | 10° | 20° | 40° |
| OZ | 0° | 10° | 20° | 30° | 40° |
| AZ | 0° | 30° | 60° | 90° | 120° |
| LAI | 5 | -50% | -25% | +25% | +50% |
| SRAD | 800 | -50% | -25% | +25% | +50% |
| LID (LIDF$_a$, LIDF$_b$) | Spherical | Uniform | Erectophile | Planophile | Plagiophile |
| Other parameters | Table A1 | -50% | -25% | +25% | +50% |


**Figure B1.** The impact of $r_{oz/sz}$ and $r_{sd}$ on SIF calculations ($F$ in Eq. (1)) when changing each input parameter in Table A1. The impact was estimated using SCOPE by changing input parameters as shown in the upper table. The difference of 0 indicates that the simulation results from LUT and SCOPE are identical when parameters that are not used in LUT.


## Appendix C: Relationship between $C_{ab}$ and $r_{756}$

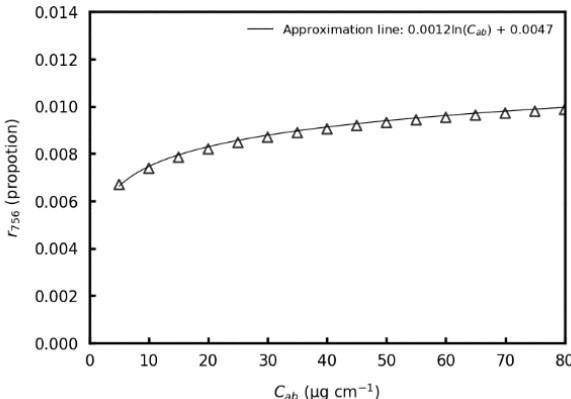

**Figure C1.** The relationship between $C_{ab}$ and $r_{756}$ computed with the SCOPE model. $r_{756}$ was simulated by correcting $C_{ab}$ to fall in the range of 5 to 80 μg cm$^{-2}$ at 5 μg cm$^{-2}$ intervals and LAI to fall in the range of 1 to 8 at an interval of 1 (open triangles).

## Appendix D: Comparison of SIF simulated using observation angles and fixed angles.

We compared the SIF simulated using observation angles and nadir angles (OZ=0) (Fig. D1). Although some models employ approaches that calculate SIF considering the observation direction, SIF simulations and validations are conducted with the angle fixed to the nadir direction for comparison with OCO-2 SIF (Bacour et al., 2019; Li et al., 2022). The validation of SIF simulations with different angles for each observation point has not been reported, and the assessment of geometric effects using different observation angles is required. Simulations using observation angles and nadir angles showed a bias, with SIF in nadir direction differing from SIF in observation direction by a maximum of +63% and a minimum of -27%. These differences vary depending on the angles between the incoming sunlight and the observation direction (the GOSAT and the nadir viewing direction). Fixing or omitting observation angles introduces uncertainties in SIF calculations; therefore, for satellites such as GOSAT with varying observation angles, it is necessary to appropriately account for geometric effects to ensure accurate simulations and analyses.

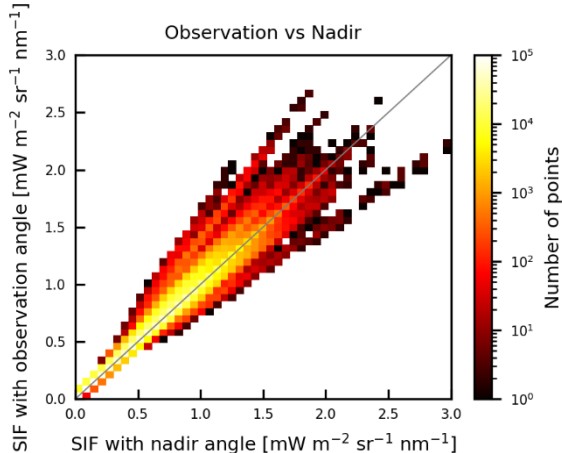

**Figure D1.** Comparison of SIF simulated by using observation angle and nadir angle. The scatter plot represents SIF at each observation point from 2009 to 2015.

**Appendix E: IDs in 42 subcontinental regions**

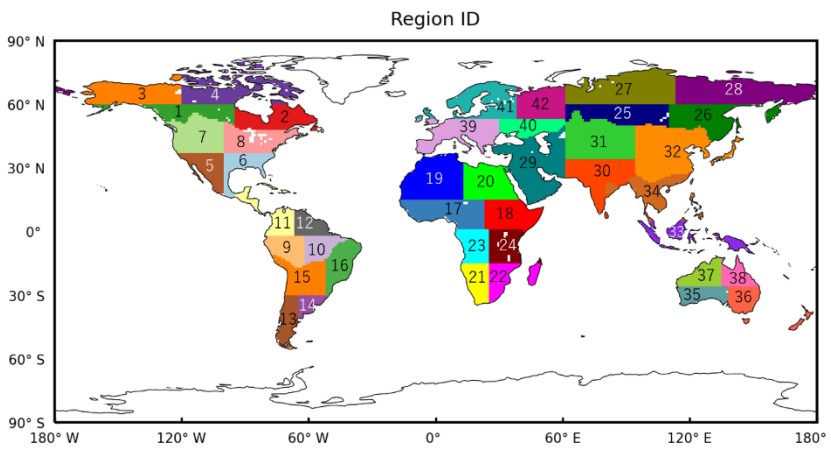


**Figure E1.** The 42 subcontinental regions.

**Code and data availability**

The model code used in this study is archived at https://doi.org/10.5281/zenodo.11243578 (Miyauchi et al., 2024)


**Author contributions**

TMiyauchi and MS conceived the ideas and designed the study. TMiyauchi developed the VISIT-SIF model, analyzed the data, and drafted the original manuscript. MS supervised the direction of the study, drafted relevant parts of the manuscript, and supported the research financially. NMH and TK provided scientific input and edited the manuscript. AI developed the

VISIT model and contributed to the manuscript. TMatsunaga supervised the GOSAT-2 project and provided essential resources.

**Competing interests**

At least one of the (co-)authors is a member of the editorial board of Geoscientific Model Development.

**Acknowledgment**

This study was funded by the GOSAT-2 project of the National Institute for Environmental Studies (NIES) and was also partly supported by the Climate Change and Air Quality Research Program in NIES and the Digital Biosphere: Integrated Biospheric Science for Mitigating Global Environment Changes in the Grant-in-Aid for Transformative Research Areas (A), Japan Society for the Promotion of Science. The authors would like to thank Yukio Yoshida and Haruki Oshio for generously sharing the GOSAT SIF data. Processing of the GOSAT SIF retrievals was performed using the Research

Computation Facility for GOSAT-2 (RCF2), and the model simulations were completed using the NIES supercomputer (HPE, Apollo 2000).

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
