# Peer review of "Process-based Modeling of Solar-induced Chlorophyll Fluorescence with VISIT-SIF version 1.0"

_EGUsphere, 2024_

## Author Response (AR2)

**Author's response (technical corrections)**

MS title: Process-based Modeling of Solar-induced Chlorophyll Fluorescence with VISIT-SIF version 1.0 (egusphere-2024-1542)

We sincerely appreciate the topic editor, Sato and reviewers for their careful reviews and comments. Below, we provided our responses to the review comments. The review comments are highlighted in orange, and our replies are presented in black.

In addition to the responses to the reviewer's comments, we revised the redundant and unclear parts of our manuscript again.
* * *
The revised manuscript improved a lot in clarity, and the authors have addressed my concerns. I think the manuscript can be accepted. Below are some minor comments:

L151 in track changed version: I think APAR shouldn't be the direct product of the terms, you probably mean "is a function of".

Reply1: We sincerely appreciate your careful review. Your comment is correct, and we agree with it. We revised the sentence as below:

L144-146:

"Here, APAR is the photosynthetically active radiation absorbed by canopy, which is a function of canopy-top irradiance, canopy reflectance, LAI and an attenuation coefficient that is a function of biome type and solar angle (Ito and Oikawa, 2002)."

L285 "data points": "data product" might be more appropriate

Reply2: We revised it as you pointed out.

L276-286:

"According to this comparison, the two data products are correlated (correlation coefficient R = 0.76; root mean squared error RMSE = 0.29 mW m-2 sr-1 nm-1) and follow the 1:1 line, indicating similar intensities."

L507: It is true that there is no great product of leaf angle at the global scale. But models like CLM

uses a PFT specific leaf angle parameterization. In the future, VISIT can probably consider similar approach to improve the simulations as leaf angle can have large impact on SIF and other simulations.

Reply3: We appreciate your valuable and insightful comment. As you pointed out, the PFT specific leaf angle parameterization is important because leaf angle has a large impact on simulations. We are currently testing LID parameter setting for each PFT. We are going to consider the same approach in CLM5 for our future model improvement.

Others:
Redundant and unclear parts of the manuscript were revised again (Discussion, Appendix B and D).

----Responses to RC1 and 2---------------------------------------------

**Author responses to comments RC1**

MS title: Process-based Modeling of Solar-induced Chlorophyll Fluorescence with VISIT-SIF version 1.0 (egusphere-2024-1542)

We sincerely thank the topic editor, Sato, reviewer1, and reviewer2 for their careful reviews and comments.
Below, we provided our responses to the review comments. The review comments are highlighted in orange, and our replies are presented in black.
* * *
Miyauchi et al describe the VISIT-SIF model for simulation of satellite SIF observations. The model was used to simulate GOSAT SIF measurements, and good results were obtained. The incorporation of SIF simulation in LSMs like VISIT provides opportunities to constrain and improve LSMs using data assimilation, and the proposed VISIT SIF model has the advantage of being capable of simulations with various viewing angles, which is not possible in many current models. However, I have several concerns as listed below. The manuscript has the potential for publication in GMD after revision.

Major comments:
1. The comparison between the VISIT-SIF model and the SIF simulations in other LSMs could be

more accurate and more in-depth. And the advantages and unique features of VISIT-SIF can be emphasized more (Also see comment 2).

Reply1: We sincerely appreciate your valuable feedback and suggestions for our manuscript. As you suggested, the comparison with other models is important for emphasizing the advantages and unique features of this model. I revised the introduction as follows. And, as the comparison with other models' approach, we added the comparison simulation using observation angles with that using fixed angles in Appendix D.

L76-82:

"In general, strict representation of radiation transfer process is required to simulate satellite SIF retrievals, especially for the satellites having off-nadir observation angles due to their complicated geometric relationships among the incidence angle of the emission to the sensor, solar azimuth, and orientation of leaves (Zhang and Zhang, 2023). But at the same time, the resolution of satellite images has kept increasing (e.g., Vicent et al., 2016), and implementation of RTMs for the observations are computationally expensive and prohibitively time-consuming. Hence, a simplified framework of avoiding direct calculation of RTMs can be an alternative and practical approach to simulate efficiently the satellite SIF retrievals."

L85-86:

"The implementation of our approach allows for easy extension to other satellite SIF retrievals."

The statement that no model can simulate GOSAT SIF (L68) is not true according to my knowledge. Lee et al (2015) and Norton et al (2018) have specifically evaluated simulations with GOSAT or used GOSAT for data assimilation. Actually, I believe most current models (listed in Table 1 in Li et al, 2022) can be used to simulate GOSAT SIF. Instead, stating the advantage of VISIT-SIF on simulating SIF observed at different angles here could be helpful.

Lee, J. E., Berry, J. A., van der Tol, C., Yang, X., Guanter, L., Damm, A., ... & Frankenberg, C. (2015). Simulations of chlorophyll fluorescence incorporated into the Community L and Model version 4. Global change biology, 21(9), 3469-3477.

Norton, A. J., Rayner, P. J., Koffi, E. N., & Scholze, M. (2018). Assimilating solar-induced chlorophyll fluorescence into the terrestrial biosphere model BETHY-SCOPE v1. 0: model description and information content. Geoscientific Model Development, 11(4), 1517-1536.

Li, R., Lombardozzi, D., Shi, M., Frankenberg, C., Parazoo, N. C., Köhler, P., ... & Yang, X. (2022). Representation of leaf-to-canopy radiative transfer processes improves simulation of far-red solar-induced chlorophyll fluorescence in the community land model

version 5. Journal of advances in modeling earth systems, 14(3), e2021MS002747.

Reply2: We removed the phrase "no model" from the sentence as follow. As you pointed out, there are some models with the potential to simulate GOSAT SIF. In our submitted manuscript, we intended to argue that SIF simulations with different angles using point-based satellite information have not been conducted. However, the sentence was confusing.

We agreed with your suggestion about simulation using different angles and revised our manuscript (same as reply 1).

L72-73:

"GOSAT has been operated since the launch in January 2009, and of which SIF retrievals have the longest observation record of any single satellite sensor."

2. In my view, one advantage of the VISIT-SIF compared with existing models is its capability to simulate satellite SIF at arbitrary viewing direction. I would suggest putting more emphasis on this and providing more analysis using SCOPE and GOSAT data.

Reply3: For the advantage of the VISIT-SIF, it is same as the reply to the comment 1. The additional analysis using SCOPE was shown in reply 4.

The lookup table approach for roz/sz and rshade/sun should be evaluated. Factors including leaf biochemical properties, leaf angle, and atmospheric conditions would also affect these parameters. It is critical to know whether the LUTs perform well when these factors are different from what is set in Table A1. I suggest running SCOPE with these parameters being varied, and evaluating the LUT results against the truth derived from SCOPE.

Reply4: We agree that the evaluation of LUT is necessary, as you mentioned. We added the following sentence and the figure evaluating the input variables and LUT in Appendix B.

**Appendix B: The evaluation of $r_{oz/sz}$ and $r_{sd}$ in LUT and SCOPE**

As a preliminary validation for the input parameters in SCOPE, the impact of $r_{oz/sz}$ and $r_{sd}$ on $F$ in Eq. (1) when changing each input parameter in Table A1 was examined (Fig. B1). The impact on $F$ was estimated using SCOPE as the difference of $r_{oz/sz}$ and $1+r_{sd}$ in Eq. (1) and (2) compared to reference values ($r_{oz/sz} = 0.5$ and $1+r_{sd} = 1.12$). $r_{oz/sz}$ increases when SZ and OZ are close to each other,

approaching a value of 1. As an exception, $r_{oz/sz}$ can exceed 1 when SZ is close to nadir, the OZ is lower value than SZ, and the LAI is lower than 2. The value of $r_{sd}$ is $0.11\pm0.05$, and it becomes 0 when OZ and SZ are identical. The reference values were determined from LUT by fixing OZ to 0 (nadir) and setting the parameters (LAI = 5, SRAD = 800 w m$^{-2}$, SZ = 30°) close to the average values in region 39. Since OZ, SZ and LAI have a significant impact on SIF calculation, they were selected as input parameters for Eqs. (15) and (16). Under the nadir-based condition, AZ had no influence on $F$. However, the angle between the solar incident direction and the observation direction, determined by SZ, OZ and AZ have a complex impact on $r_{oz/sz}$ and $r_{sd}$, selecting AZ as an input parameter. SRAD is a fundamental input variable for radiative transfer calculations and was applied in the LUT to align the light environment with VISIT. Variables such as $w$, $H_c$, and $C_{ab}$ have a relatively large effect on SIF calculations; however, since they are not computed within VISIT, fixed values were applied to them. SRAD is a fundamental input variable for radiative transfer calculations and was applied in the LUT to align the light environment with VISIT, although it caused only a small change in $r_{oz/sz}$ and $r_{sd}$. The variables such as $w$, $H_c$, and $C_{ab}$ have a relatively large impact on SIF calculations, however, since they are not computed within VISIT, fixed values were applied to them. LID also has a significant impact, but since there is no reasonable method for setting LID in global calculations for VISIT, it was fixed to spherical. The impact of other parameters fell within ±3% and they were not selected as input parameters.

| Parameter | Reference value | ◇ | ◯ | △ | ☐ |
|---|---|---|---|---|---|
| SZ | 30° | 0° | 10° | 20° | 40° |
| OZ | 0° | 10° | 20° | 30° | 40° |
| AZ | 0° | 30° | 60° | 90° | 120° |
| LAI | 5 | -50% | -25% | +25% | +50% |
| SRAD | 800 | -50% | -25% | +25% | +50% |
| LID (LIDF$_a$, LIDF$_b$) | Spherical | Uniform | Erectophile | Planophile | Plagiophile |
| Other parameters | Table A1 | -50% | -25% | +25% | +50% |

**Figure B1.** The impact of $r_{oz/sz}$ and $r_{sd}$ on SIF calculations ($F$ in Eq. (1)) when changing each input parameter in Table A1. The impact was estimated using SCOPE by changing input parameters as shown in the upper table. The difference of 0 indicates that the simulation results from LUT and SCOPE are identical when parameters that are not used in LUT.

The evaluation of the geometry effect in Figure 3 looks a bit weird to me. The issue with most current models is that they only simulate for the nadir direction. I suggest also comparing the simulations with a scenario assuming nadir viewing angle.

Reply5: As you mentioned, many models calculated the nadir SIF, and we believe that the comparison between SIF calculation using observation angles and those using nadir angles would be meaningful. We added the comparison with the nadir SIF as additional information about the effect of considering

observation angles in Appendix D.

**Appendix D: Comparison of SIF simulated using observation angles and fixed angles.**

We compared the SIF simulated using observation angles and nadir angles (OZ=0) (Fig. D1). Although some models employ approaches that calculate SIF considering the observation direction, SIF simulations and validations are conducted with the angle fixed to the nadir direction for comparison with OCO-2 SIF (Bacour et al., 2019; Li et al., 2022). The validation of SIF simulations with different angles for each observation point has not been reported, and the assessment of geometric effects using different observation angles is required. Simulations using observation angles and nadir angles showed a bias, with SIF in nadir direction differing from SIF in observation direction by a maximum of +63% and a minimum of -27%. These differences vary depending on the angles between the incoming sunlight and the observation direction (the GOSAT and the nadir viewing direction). Fixing or omitting observation angles introduces uncertainties in SIF calculations; therefore, for satellites such as GOSAT with varying observation angles, it is necessary to appropriately account for geometric effects to ensure accurate simulations and analyses.

[Figure]

**Figure D1.** Comparison of SIF simulated by using observation angle and nadir angle. The scatter plot represents SIF at each observation point from 2009 to 2015.

3. I have a few concerns regarding the method:

a. The fluorescence yield at the leaf level and the photosystem level seem confused in the SIF model. Eq.2 requires leaf-level fluorescence yield, while Eq.3 provides photosystem-level fluorescence yield. Only part of photosystem-level fluorescence escapes the leaf as some are absorbed within leaf (Porcar-Castell et al, 2021). Using photosystem-level fluorescence yield as a leaf-level parameter would lead

to bias in the simulation. Various approaches have been used to address this issue in other models, for example, see Lee et al, (2015) and Li et al, (2022).

Porcar-Castell, A., Malenovský, Z., Magney, T., Van Wittenberghe, S., Fernández-Marín, B., Maignan, F., ... & Logan, B. (2021). Chlorophyll a fluorescence illuminates a path connecting plant molecular biology to Earth-system science. Nature plants, 7(8), 998-1009.

Lee, J. E., Berry, J. A., van der Tol, C., Yang, X., Guanter, L., Damm, A., ... & Frankenberg, C. (2015). Simulations of chlorophyll fluorescence incorporated into the Community L and Model version 4. Global change biology, 21(9), 3469-3477.

Li, R., Lombardozzi, D., Shi, M., Frankenberg, C., Parazoo, N. C., Köhler, P., ... & Yang, X. (2022). Representation of leaf-to-canopy radiative transfer processes improves simulation of far-red solar-induced chlorophyll fluorescence in the community land model version 5. Journal of advances in modeling earth systems, 14(3), e2021MS002747.

Reply6: As you pointed out, the approach of converting the fluorescence yield at the photosystem-level to the leaf level is important. In our study, we used SCOPE in a simplified manner to calculate $r_{oz/sz}$, $r_{sd}$, $f_u$, and $r_{756}$, and compute canopy-level SIF considering the radiative transfer process. However, we have not accounted for internal reabsorption within a single leaf. The reabsorption by chlorophyll is only considered in the calculation of $r_{756}$. Therefore, VISIT-SIF remains uncertainty in the fluorescence emission at the leaf level. The conversion from photosystem level to leaf level fluorescence yield remains is a challenge for our future model improvements. Regarding these points, we revised the manuscript as follows.

L154-156(method):
"The variable can be used to estimate canopy-level fluorescence by considering radiative transfer processes within the canopy layers; however, the reabsorption at single-leaf-level in fluorescence yield was not reflected."

L399-407(discussion):
"By using SCOPE in a simplified manner, the radiative transfer process between canopy layers to calculate SIF in arbitrary observation direction can be computed; however, uncertainty remains in the calculation of chlorophyll fluorescence yield at leaf- and canopy-level. While the reabsorption by chlorophyll was considered in the calculation of $r_{756}$, the internal reabsorption at the single-leaf-level was not directly considered in the fluorescence yield in Eq. (2). Li et al. (2022) used the simplified scattering fluorescence coefficient in SCOPE, which estimates the chlorophyll fluorescence from the excitation light, to convert the fluorescence yield at the photosystem level to the leaf level. However, since VISIT-SIF and SCOPE differ in the model structure of canopy layers and spectral calculations, a conversion approach for fluorescence yield at leaf level has not been implemented in VISIT-SIF. The conversion of fluorescence yield from photosystem level to leaf-level is a challenge for future

improvements of VISIT-SIF."

b. L131: I suggest providing some details on how VISIT simulates APAR, it is confusing to me why that can be considered APARsun.

Reply7: We added the sentences about $APAR_{sun}$ calculation as bellow.

L147-152:

"Here, APAR is the photosynthetically active radiation absorbed by canopy, which is the product of canopy-top irradiance, canopy reflectance, LAI and an attenuation coefficient that is a function of biome type and solar angle (Ito and Oikawa, 2002). $APAR_{sun}$ consists of the absorbed beam, diffuse, and scattered beam with the sunlit layer based on Farquhar model (de Pury and Farquhar 1997). Under the assumption that VISIT simulates biochemical processes occurring within sunlit leaves where the viewing angle coincides with the sun zenith angle, we adopted $APAR_{sun}$ in Eq. (2)."

c. L132: How is fu derived?

Reply8: $f_u$ was calculated as the average fraction of upward chlorophyll fluorescence to tatal emitted chlorophyll fluorescence from photosystem across the 60 canopy layers in SCOPE. We revised the sentences related to $f_u$ as follow.

L152-157:

"The variable $f_u$ represents the fraction of the SIF emitted in the upward direction to that in both the upward and downward directions at the canopy-level. It was obtained as the average fraction across the 60 canopy layers by operating the SCOPE model. The variable can be used to estimate canopy-level fluorescence by considering radiative transfer processes within the canopy layers; however, reabsorption at single leaf level in fluorescence yield was not reflected. In addition, Equation (2) describes indirect incorporation of VISIT and SCOPE by multiplying $\Phi_{F,sun}$, $r_{oz/sz}$ and $f_u$."

Minor comments:

L16: We also found … This sentence is not clear to me, please rephrase.
L40: "close to the oxygen absorption band" might be a more accurate description

Reply9: We revised as follows.

L20-21:

We also found that the mean seasonal variability in the simulated SIFs was closely consistent with the GOSAT SIF retrievals at the subcontinental scale.

L43:

in the oxygen absorption bands between 756 and 759 nm (Oshio et al., 2019) and between 734 and 758 nm (Joiner et al., 2013)

Others:
1.  We added the caption for Fig. 5c.
2.  We had revised the Eq.14 due to a mistake in unit.

**Author responses to comments RC2**

MS title: Process-based Modeling of Solar-induced Chlorophyll Fluorescence with VISIT-SIF version 1.0 (egusphere-2024-1542)

We sincerely thank the topic editor, Sato, reviewer1, and reviewer2 for their careful review and comments.

Below, we provided our responses to the review comments. The review comments are highlighted in orange, and our replies are presented in black.
* * *
Miyauchi et al. developed a new model, VISIT-SIF version 1.0, to predict SIF and compared it with GOSAT SIF retrievals. The VISIT-SIF model is a significant contribution to the modeling of SIF and has the potential to further our understanding of carbon dynamics. The manuscript is well written and suitable for publication in GMD. However, I have several comments listed below. A revision is necessary before publication.

Major comments:

1. The abstract could be strengthened by emphasizing the uniqueness of the VISIT-SIF model and mentioning how it differs from other existing models. The authors could highlight the capability of the VISIT-SIF model to simulate SIF from different angles. Additionally, it would be beneficial to briefly summarize the application of modeling SIF to broader contexts, either in the abstract or conclusion.

Reply1: Thank you for your careful and helpful review. We revised the abstract following your comments.

L13-18:

"Implementation of radiation transfer models (RTMs) helps to address the interaction of chlorophyll fluorescence with vegetation and atmosphere. However, the computation of RTMs becomes more time-consuming, which can make it impractical in application to satellite observations with larger data volumes. This study resolves this issue by parameterizing the radiation transfer processes and its geometric relationships. This approach enables ease of implementation of VISIT-SIF for simulating satellite SIF retrievals even for the satellites having off-nadir observation angles."

2. GOSAT-SIF data used in this study contain negative values. Why do the authors not consider data filtering on the GOSAT-SIF before comparison with VISIT-SIF?

Reply2: As you mentioned, the negative SIF values are not actual physical quantities caused by retrieval nose and such nosy data often removed by filtering. However, in this study, we used all satellite observation data for comparisons to prevent significant biases in the probability distribution of SIF variability. Relevant descriptions were described as below.

L313-315:

"The negative SIF values are not actual physical quantities because of the presence of retrieval noise, but this study used all the satellite observations without discarding the negative values to prevent significant biases in the probability distribution of the SIF variability."

Reply3: $f_u$ was calculated as the average fraction of upward chlorophyll fluorescence to the total emitted chlorophyll fluorescence across the 60 canopy layers in SCOPE. We revised the sentences related to $f_u$ as follow.

L152-157:

"The variable $f_u$ represents the fraction of the SIF emitted in the upward direction to that in both the upward and downward directions at the canopy level. It was obtained as the average fraction across the 60 canopy layers by operating the SCOPE model. The variable can be used to estimate canopy-level fluorescence by considering radiative transfer processes within the canopy layers; however, reabsorption at single leaf level in fluorescence yield was not reflected. In addition, Equation (2) describes indirect incorporation of VISIT and SCOPE by multiplying $\Phi_{F,sun}$, $r_{oz/sz}$ and $f_u$."

Minor comments:
Line 32: "increases to prevent damae to the photosynthetic system due to the accumulation of excess energy." Should damae be damage?

Reply4: We appreciate your careful review and pointing out the misspellings. As you mentioned, it was a misspelling. We corrected it to damage.

Line 33-34: "hence, the quantum yield of photochemistry is positively and negatively correlated with fluorescence and heat dissipation" to "hence, the quantum yield of photochemistry is positively correlated with fluorescence and negatively correlated with heat dissipation".

Line 69: "has not been developed since the launch of GOSAT in January 2009" to "has not been developed until the launch of GOSAT in January 2009".

Line 257: "for the satellite observations and model simulations" to "between the satellite observations and model simulations".

Ling 258: "according to this comparison" to "According to this comparison"

Line 377: "temperature and water and light limitations." To "temperature, and water and light limitations."

Line 401-403: Please give relevant reference to GOME-2, OCO-2, TROPOMI, etc.

Reply5: We appreciate your careful reading. As you pointed out, we revised our manuscript.

Regarding L33-34, since the positive or negative correlation among photochemistry, heat dissipation, and chlorophyll fluorescence is not necessarily consistent under varying environmental conditions, we revised it as follow.

L36-37:

"- hence, the quantum yield of photochemistry is correlated with fluorescence and heat dissipation (Flexas et al., 2000)."

Regarding L33-34, we changed the sentence according to RC1.

L72-73:

"GOSAT has been operated since the launch in January 2009, and of which SIF retrievals have the longest observation record of any single satellite sensor."

Figure 1: The description to the figure is too short. Please give necessary information about the diagram, for example the major components or the work flows.

Reply6: We added the explanation of our model and work flows as follow.

L127-132:

"The model system consists of biochemical/biophysical processes and geometric and radiative transfer processes. The former simulates the canopy structures and the radiative conditions within the canopy and the actual and potential electron transport rates for a given grid. The simulated electron transport

rates are inputs for the quantum yield of chlorophyll fluorescence, and the absorbed photosynthetically active radiation (APAR) is used to calculate SIF. The latter simulates radiative transfer processes for the SIF emitted from the upper canopy. The practical operation manner to simplify the simulation of radiation transfer processes is given later in this subsection."

Figure 6: The circles are overlapped. It is recommended to show them in different colors or shapes.

Reply7: We revised Figure 6 using different colors.

[Figure]

**Figure 6.** The relationships between $R^2$ and RMSE for the mean monthly variability in the VISIT-SIF simulations and GOSAT retrievals in the 42 subcontinental regions. The numbers in the figure correspond to the regional IDs shown in Fig. C1.

Figure 7: Only 9 regions are selected to show, is there any specific reason that other regions are not shown?

Reply8:
We selected nine characteristic regions for discussion. There is no specific reason for not displaying other regions; however, Figure 7 presents characteristic all-period monthly SIF extracted from 7-years mean monthly SIF shown in Figure 6. To avoid redundancy, other regions were not displayed.

Others:

3.  We added the caption for Fig. 5c.

4.  We had revised the Eq.14 due to a mistake in unit.